# Fast-YOLO Network Model for X-Ray Image Detection of Pneumonia

**Bin Zhao** [1,2,3,†] 🆔, **Lianjun Chang** [1,3,*,†] **and Zhenyu Liu** [1,*]

1 Information Science and Engineering School, Shenyang University of Technology, Shenyang 110870, China; zhaobin@stumail.neu.edu.cn
2 College of Information Science and Engineering, Northeastern University, Shenyang 110819, China
3 SIASUN Robot & Automation Co., Ltd., Shenyang 110169, China
* Correspondence: changlianjun@smail.sut.edu.cn (L.C.); liuzhenyu@sut.edu.cn (Z.L.)
† These authors contributed equally to this work.

**Abstract:** Pneumonia is a respiratory infection that affects the lungs. The symptoms of viral and bacterial pneumonia are similar. In order to improve automatic detection efficiency regarding X-ray images of pneumonia, this paper, we propose a novel pneumonia detection method based on the Fast-YOLO network model. First, we re-annotated the open-source dataset of MIMIC Chest X-ray pneumonia, enhancing the model's adaptability to complex scenes by incorporating Mixup, Mosaic, and Copy–Paste augmentation methods. Additionally, CutMix and Random Erasing were introduced to increase data diversity. Next, we developed a lightweight FASPA Fast Pyramid Attention Mechanism and designed the Fast-YOLO network based on this mechanism to effectively address the complex features in pneumonia X-ray images, such as low contrast and an uneven distribution of local lesions. The Fast-YOLO network improves upon the YOLOv11 architecture by replacing the C3k2 module with the FASPA attention mechanism, significantly reducing the network's parameter count while maintaining detection performance. Furthermore, the Fast-YOLO network enhances feature extraction capabilities when handling scenes with geometric deformations, multi-scale features, and dynamic changes. It expands the receptive field, thereby balancing computational efficiency and accuracy. Finally, the experimental results demonstrate that the Fast-YOLO network, compared to traditional convolutional neural network methods, can effectively identify pneumonia regions and localize lesions in pneumonia X-ray image detection tasks, achieving significant improvements in FPS, precision, recall, mAP @0.5, and mAP @0.5:0.95. This confirms that Fast-YOLO strikes a balance between computational efficiency and accuracy. The network's excellent generalization capability across different datasets has been validated, showing the potential to accelerate the pneumonia diagnostic process for clinicians and enhance diagnostic accuracy.

**Keywords:** classification detection; pneumonia detection; model optimization; deep learning

## 1. Introduction

Early diagnosis of pneumonia is of great significance in disease treatment and prognosis evaluation, especially in regard to imaging detection. Computed tomography (CT) and X-ray technology have been widely used to detect and diagnose pneumonia [1,2]. X-ray images can provide detailed anatomical structures of the lungs and high-resolution images of diseased areas, providing clinicians with critical auxiliary information. As an inflammatory lung disease with multiple causes, the clinical manifestations of pneumonia are highly heterogeneous, with imaging features often being in low contrast and showing

an uneven lesion distribution, hampering work involving traditional artificial diagnosis methods. Manual diagnosis usually takes a long time, and the result of the diagnosis is subject to the corresponding doctor's experience level, increasing the misdiagnosis rate and the inconsistency of diagnosis results. During a pneumonia epidemic, the diagnostic process becomes time-consuming and error-prone as the number of patients surges and clinicians are under pressure to review large quantities of imaging data. In recent years, with the rapid development of medical imaging technology and artificial intelligence, the medical image analysis method based on deep learning now provides a new possibility for the early automatic diagnosis of pneumonia [3]. In particular, convolutional neural networks have allowed remarkable progress in medical image analysis, especially in automated pneumonia detection. The current mainstream target detection algorithms can achieve efficient and accurate lesion detection in cases involving complex backgrounds through automatic feature extraction and precise positioning. Among these methods, designed with the innovative idea of transforming the object detection task into a single regression problem, the YOLO model can realize the synchronous prediction of an object's position and category with very high detection efficiency and calculation speed. This gives the YOLO model a significant advantage when analyzing X-ray image data. An automatic diagnosis system based on deep learning can improve diagnostic efficiency and reduce the misdiagnosis rate to a certain extent, providing more reliable decision support for clinicians. Further optimization of the detection accuracy and robustness of the YOLO model in complex environments, combined with the characteristics of medical imaging, is expected to promote the development of the automated diagnosis of pneumonia X-ray images and the intelligent transformation of medical imaging diagnosis from being experience-driven to data-driven. Significant progress has been made in research related to pneumonia detection based on X-ray images. Prasath J et al. [4] proposed an optimized dual transformer residual super-resolution network (DTRSN-XRI-CPI) for identifying pneumonia in chest X-ray images, extracting image features such as color, shape, spatial, texture, and relationships. The model improved performance metrics such as accuracy, recall, and F1-score compared to existing intelligent computational frameworks. Rana N et al. [5] proposed a pneumonia detection model based on chest X-ray data to achieve early and efficient disease detection using advanced data analysis techniques. Their study introduced an unsupervised learning-based solution to address data scarcity and privacy concerns. The training data for this model were sourced from multiple healthcare institutions, covering chest X-ray images from both pneumonia patients and healthy individuals. The proposed model outperformed existing pneumonia detection models in terms of performance. Chen Q et al. [6] presented a mixed-scale dynamic attention transformer aided by large language models (LLMs) for automatic pediatric pneumonia diagnosis. Evaluations of pediatric chest X-ray datasets, including Pneumonia Physician, Guangzhou Women and Children's Medical Center, and NIH CXR14, showed that this method outperformed rival methods in key metrics such as accuracy, AUC, precision, recall, and F1 score, demonstrating its potential for pediatric pneumonia imaging. Zhou T et al. [7] introduced a computer-aided diagnosis model, ResFormer (Identity-mapping ResFormer), for pneumonia X-ray images. This module integrates gradient features at different stages using transformer operations. The model was validated using a lung X-ray dataset, confirming that ResFormer can effectively assist doctors in making efficient and accurate pneumonia diagnoses. While related deep learning algorithms perform well in pneumonia detection tasks, challenges remain in identifying small lesions and pneumonia manifestations with complex shapes due to significant variations in lesion size, shape, and distribution. Noise and low-contrast lesions in X-ray images can affect the detection accuracy of models.

To address these issues, in this paper, we propose an optimized detection model based on FAST-YOLO to improve lesions' recognition accuracy and localization precision in pneumonia X-ray images. The FAST-YOLO model effectively preserves feature expression capabilities while significantly reducing computational complexity. This enhancement improves feature extraction abilities and expands the receptive field [8–12]. The Fast-YOLO network improves upon the YOLOv11 architecture by replacing the C3k2 module with the FASPA attention mechanism, significantly reducing the network's parameter count while maintaining detection performance. This model overcomes the limitations of traditional models in pneumonia detection, balancing detection speed with computational resource requirements. The FAST-YOLO algorithm framework optimizes the detection efficiency of the YOLO model for pneumonia X-ray images, improving its detection performance and practical application value. The results of this study indicate that the pneumonia X-ray image detection method based on the FAST-YOLO model can provide clinicians with efficient and accurate auxiliary tools, helping to accelerate the pneumonia diagnosis process and improve diagnostic accuracy.

To access the open-source MIMIC Chest X-ray pneumonia dataset, visit the following address:

https://physionet.org/content/mimic-cxr/2.0.0/ (accessed on 1 January 2024).

## 2. Pneumonia Dataset and Evaluation Metrics

### 2.1. Pneumonia Dataset

X-ray images of pneumonia typically exhibit complexity, diversity, and low contrast, necessitating the construction of an efficient and accurate pneumonia X-ray image dataset [13–15]. We used the open-source MIMIC Chest X-ray pneumonia dataset; the images were re-annotated using the LabelImg tool (as detailed in Table 1), which includes annotations for five categories: bacterial pneumonia, viral pneumonia, illness, healthy, and tuberculosis.

**Table 1.** Lung condition categories and labels.

| Category of Labels | Number of Labels |
| --- | --- |
| Pneumonia Bacteria | 987 |
| Pneumonia Virus | 895 |
| Sick | 812 |
| Healthy | 875 |
| Tuberculosis | 625 |

In this paper, we propose a targeted online data augmentation method. This method integrates several advanced image enhancement techniques, including Mixup and Mosaic, for comprehensive dataset preprocessing, significantly improving the model's generalization ability and robustness [16–19].Annotated examples from the pneumonia dataset are shown in Figure 1. Specifically, the data augmentation process involved HSV transformation, translation, scaling, horizontal flipping, random cropping, zooming, and stitching [20,21]. These techniques not only effectively increase the sample size but also enhance the model's sensitivity and adaptability in small-object detection, providing strong support for the overall optimization of model performance.

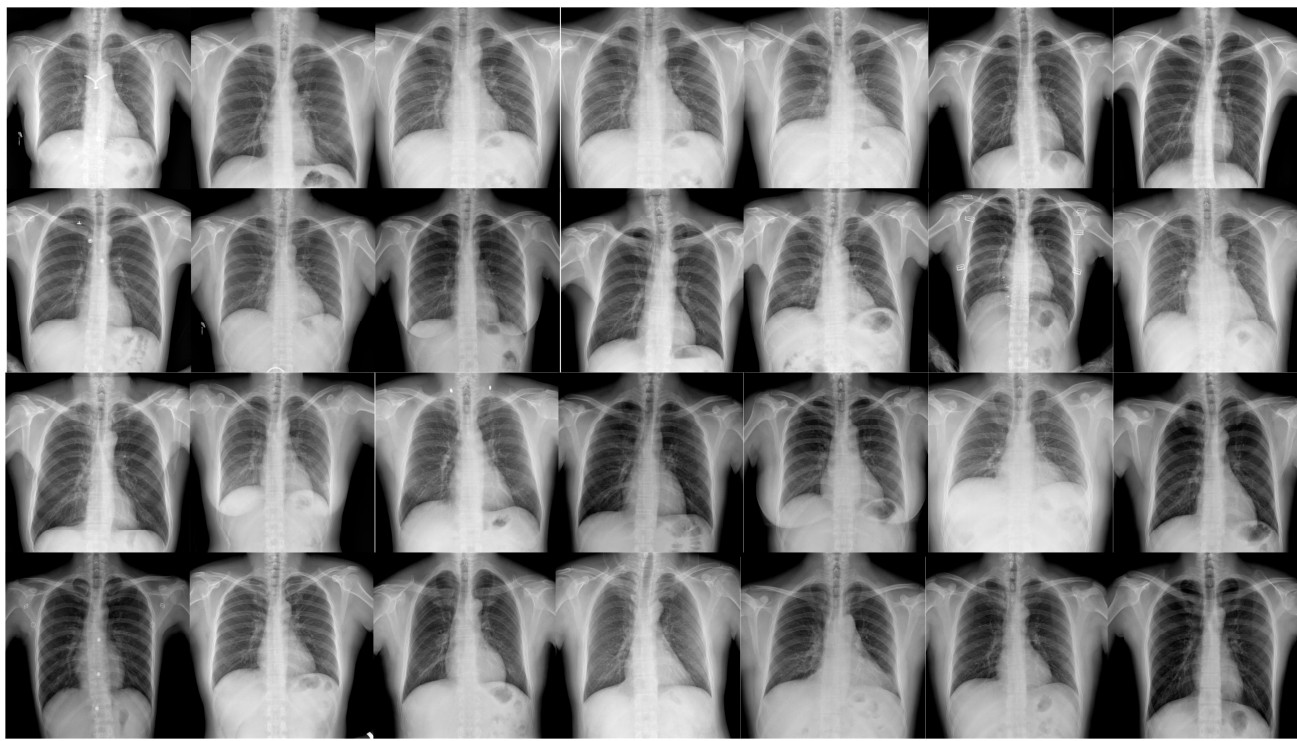

**Figure 1.** Pneumonia detection dataset.

*2.2. Loss Function and Evaluation Metrics*

A pneumonia X-ray image detection system should not only achieve high accuracy in lesion detection but also rely on a scientifically designed loss function and evaluation metrics to optimize model performance. In deep learning, the loss function and evaluation metrics are two indispensable core components in model training and evaluation. 1. The loss function defines how a model adjusts its parameters during the training process to minimize prediction errors or maximize a given objective. It directly influences the calculation of gradients and the updating of parameters [22–24]. The goal of deep learning models is typically to optimize a model's predictive performance by minimizing the loss function. 2. Evaluation metrics are used to assess a model's performance outside of training, helping users understand the model's actual performance across different tasks. Metrics such as accuracy, precision, recall, and F1 score are often used, especially in cases of class imbalance, as they provide richer information than simple loss functions. A model can be evaluated at different stages of training, and adjustments to hyperparameters or training strategies can be made based on the evaluation metrics, a common method for improving model performance. The loss function forms the foundation of the optimization process, directly guiding how a model adjusts its parameters, while evaluation metrics serve as the standard for assessing a model's final performance. Both must work in tandem to achieve optimal performance. The loss function for object detection algorithms typically includes the following components:

(1) Localization loss: This is used to measure the error between the predicted bounding box and the actual ground-truth bounding box. Mean squared error (MSE) is generally used to compute the four parameters of the bounding box: the center coordinates, $x_i$ and $y_i$; width, $w_i$; and height, $h_i$. The loss function is as follows:

$$L_{\text{loc}} = \sum_i \lambda_{\text{coord}} \left[ 1_{\text{obj}} \left( (x_i - \hat{x}_i)^2 + (y_i - \hat{y}_i)^2 + (w_i - \hat{w}_i)^2 + \left(h_i - \hat{h}_i\right)^2 \right) \right] \quad (1)$$

where $\lambda_{\text{coord}}$ is a weight factor, ensuring that the loss is computed only when a ground-truth box exists. $1_{\text{obj}}$ is an indicator function, which takes the value of 1 when the sample contains the object.

(2) Confidence loss: Confidence loss measures the certainty of whether an object is present in the predicted box. The core of this loss lies in evaluating the difference in confidence between the predicted and ground-truth boxes. Binary cross-entropy loss is commonly used to compute this, optimizing the model's ability to predict confidence in object detection tasks. By simultaneously optimizing the confidence loss for both object and background boxes, a model can effectively balance the impact of positive and negative samples, enhancing the accuracy and robustness of the detection results. This design is crucial for handling multi-object detection tasks in complex scenes and helps improve a model's ability to differentiate object boundaries and categories. The loss function is as follows:

$$L_{\text{conf}} = \sum_i 1_{\text{obj}} \left(C_i - \hat{C}_i\right)^2 + \lambda_{\text{noobj}} 1_{\text{noobj}} \left(C_i - \hat{C}_i\right)^2 \qquad (2)$$

where $C_i$ is the confidence of the predicted box and $\hat{C}_i$ is the confidence of the ground-truth box. $1_{\text{noobj}}$ is an indicator function, which takes the value of 1 when the sample does not contain the object.

(3) Classification loss: The YOLO series uses cross-entropy loss to compute classification errors, measuring the difference between the predicted and actual categories. Each detection box has a category label, and the YOLO model computes a probability distribution for each box. The loss function is as follows:

$$L_{\text{cls}} = \sum_i 1_{\text{obj}} \sum_c \left(P_{i,c} - \hat{P}_{i,c}\right)^2 \qquad (3)$$

where $P_{i,c}$ is the predicted probability for the $i$-th box belonging to class $c$, and $\hat{P}_{i,c}$ is the actual probability of the class.

(4) Total loss function: The total loss function in YOLO is typically the weighted sum of the above three losses:

$$L_{\text{total}} = \lambda_{\text{loc}} L_{\text{loc}} + \lambda_{\text{conf}} L_{\text{conf}} + \lambda_{\text{class}} L_{\text{class}} \qquad (4)$$

In this formula, weighted factors, $\lambda_{\text{loc}}$, $\lambda_{\text{conf}}$, $\lambda_{\text{class}}$, are included to balance the impact of each component.

To accurately evaluate the model's robustness and lesion detection precision, we employed six main evaluation metrics, namely, precision, recall, F1 score, mean average precision (mAP), and frames per second (FPS), for performance assessment.

$$\begin{cases} \text{Precision} = TP/(TP + FP) \times 100\% \\ \text{Recall} = TP/(TP + FN) \times 100\% \\ F1 \text{ score} = \frac{2 \times \text{Precision} \times \text{Recall}}{\text{Precision} + \text{Recall}} \\ mAP = \frac{-\sum_{i=1}^{n} \int_0^1 P_i(R)\mathrm{d}R}{N} \\ FPS = \text{FigureNumber}/\text{TotalTime} \end{cases} \qquad (5)$$

In this context, TP (true positive) refers to the number of actual positive samples correctly predicted by the model, FP (false positive) refers to the number of actual negative samples incorrectly predicted to be positive, and FN (false negative) indicates the number of actual positive samples incorrectly predicted to be negative by the model. $N$ represents the total number of sample categories, while $P_i(R)$ denotes the precision at a specific recall rate (recall) for the $i$-th class. $AP_i$ (average precision) represents the average precision for the $i$

th class, which is used to evaluate the detection performance of that class. FigureNumber indicates the total number of processed images, which is a key parameter in the evaluation process. TotalTime refers to the time required to process all images.

By effectively combining the above loss function components and integrating efficient evaluation metrics (such as mAP, IoU, a precision–recall curve, etc.), the pneumonia X-ray image detection system can achieve high detection accuracy while maintaining robustness and reliability, thereby providing reliable auxiliary diagnostic support for clinical applications.

# 3. FAST-YOLO Network

## 3.1. FASPA Attention

The FASPA (FAST Atrous Spatial Pyramid Attention) module applies RepConv to the gradient flow branch to improve both feature extraction and gradient flow efficiency. The size of FASPA can be adjusted by using a scaling factor, *n*, enabling it to support both small and large model architectures. The structure of the FASPA module comprises several convolutional layers, designed to extract, transform, and fuse features to perform specific computational tasks, as shown in Figure 2. The FASPA module's design includes components such as 1×1 convolution, split operation, 3×3 convolution, and RepConv modules.

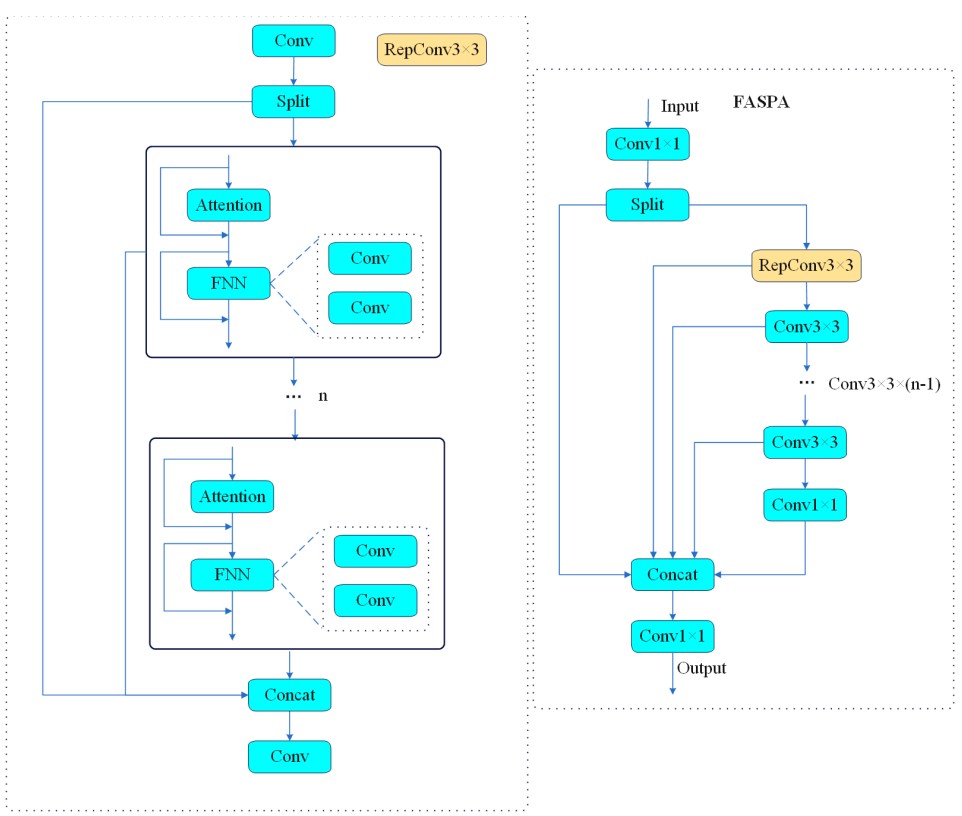

**Figure 2.** Diagram of the structure of the FASPA module.

The specific components of the FASPA module are as follows:

(1)  A Conv1×1 layer and a Conv3×3 layer: The Conv1×1 convolutional layer is used to adjust the dimensionality of the input feature map through linear transformations between channels, serving the purpose of either compressing or expanding the dimensionality of the input data. Specifically, the Conv1×1 convolution operates

by modifying the number of channels in the feature map while keeping the spatial dimensions unchanged.

A Conv3×3 convolutional layer is a commonly used kernel size in CNNs. It performs convolution operations in local regions of the input feature map using a sliding window. These layers are effective in capturing spatial local features and computing convolutions at each position in the feature map. The formula for the convolution layer is as follows:

$$Y_{ij} = \sum_{m=0}^{k-1} \sum_{n=0}^{k-1} X_{i+m,j+n} \cdot W_{m,n} + b \tag{6}$$

where $X$ is the input feature map, $W$ is the convolution kernel, $b$ is the biased term, $Y$ is the output feature map, and $k$ is the size of the convolution kernel.

(2) Split layer: After the Conv1×1 convolution, the input feature map undergoes a splitting operation. This operation divides the input feature map into multiple sections, with each section passed to different processing paths. Specifically, the input is split into several branches, and each branch performs distinct convolution operations.

(3) RepConv3×3 layer: The core idea of RepConv is to use multi-branch convolution layers during training and then re-align the parameters of each branch to the main branch during inferencing. This significantly reduces computational load and memory consumption. The RepConv method is designed to optimize the efficiency of standard convolutional layers by decreasing computational overhead and improving computational efficiency or by enhancing feature learning in a more efficient manner. RepConv not only reduces computational burden but also enhances operational efficiency without sacrificing the model's expressive capacity.

(4) Concat layer: After all branches complete their convolution operations, the feature maps from each branch are concatenated. The concatenation operation generally connects the feature maps from multiple branches along the channel dimension, thereby enhancing the network's feature representation ability. This operation facilitates the fusion of features extracted through different convolution paths. The concatenated feature map is then passed through a 1×1 convolutional layer to yield the final output. The purpose of this convolutional layer is to further adjust the output dimensions via linear transformations, making the output compatible with subsequent network layers or target outputs.

This module processes the input data via a multi-path structure, where each path extracts features through different convolution operations. The features from all paths are fused through concatenation and passed through a 1×1 convolutional layer for the final output. This structure effectively enhances the diversity and expressiveness of feature extraction, thereby improving the model's performance.

### 3.2. Network Architecture

The core of the Fast-YOLO network continues to follow the single-stage detection approach of the YOLO series, consisting of two main components, namely, a Backbone network and a Head network, as shown in Figure 3. By replacing the C3k2 module with the FASPA attention mechanism, the network effectively reduces computational complexity while maintaining the expressive capability of the features.

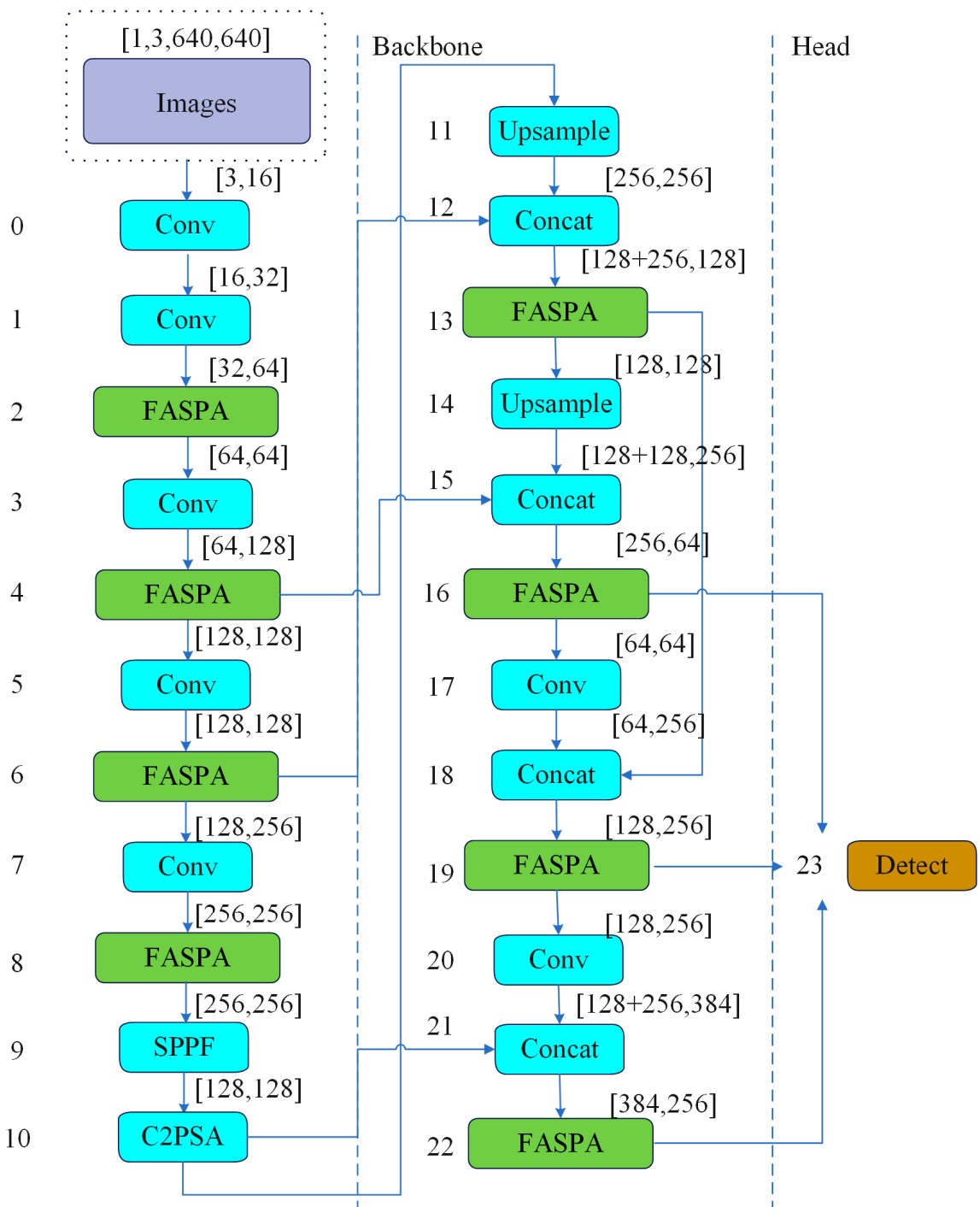

**Figure 3.** Structure of the FAST-YOLO based on FASPA.

(1)  Backbone: The structure of the Backbone primarily consists of the Conv, FASPA, SPPF, and C2PSA modules. The SPPF (Spatial Pyramid Pooling Field) module enhances feature extraction in convolutional neural networks, particularly when processing input images of varying sizes, improving a network's adaptability and performance. The C2PSA module optimizes the model's effectiveness in handling multi-scale features, supporting an optional residual structure that facilitates gradient propagation and enhances the accuracy and robustness of object detection. The convolutional module automatically learns local features from the input data, passing them through successive layers while preserving essential information through convolutional and pooling layers. The FASPA attention introduces the PSA (Pyramid Spatial Attention)

mechanism, enhancing the model's feature extraction capability and increasing the precision of attention focusing. PSA performs excellently in handling multi-scale features, contributing to improving object detection accuracy and robustness. Additionally, FASPA attention supports an optional residual structure that optimizes gradient propagation, thereby improving network training efficiency and stability. This design effectively helps a model to capture complex nonlinear relationships between input features, enriching feature representations, enhancing expressive capacity, and creating a multi-scale representation akin to a feature pyramid.

(2) Head: The structure of the Head primarily consists of the Upsample, Concat, and FASPA modules. The primary function of the Upsample module in deep learning networks is to upsample the spatial resolution of feature maps, thereby restoring an image to its original or near-original size. The Head is mainly responsible for making the final regression predictions, utilizing feature maps extracted by the Backbone network to detect object bounding boxes and classify their categories. The Head network combines the Generalized Intersection over Union (GIoU) loss function and the Weighted Non-Maximum Suppression (NMS) technique, optimizing the localization accuracy of bounding boxes and the accuracy of category predictions. The Head network is adaptable to detecting objects of various sizes within an image, thereby enhancing the model's ability to operate effectively in complex scenarios.

(3) Data Augmentation: Mixup, Mosaic, and Copy–Paste augmentation techniques enhance a model's adaptability to complex scenes. Additionally, the introduction of CutMix and Random Erasing simulates target detection under occlusion scenarios.

## 4. Results

The operating environment of the experimental server is shown in Table 2:

**Table 2.** Server operating environment.

| Name | Version |
| --- | --- |
| OS | Ubuntu MATE 16.04 |
| CPU | Intel(R) Xeon(R) CPU E5-2620 v4 @ 2.10 GHz |
| RAM | 128 GB |
| GPU | GeForce RTX 3090$\times$2 |
| Driver | 455.23.05 |
| CUDA | 11.1 |
| python | 3.7.13 |
| torch | 1.10.1+cu11 |
| torchvision | 0.11.2++cu111 |

### 4.1. Pneumonia X-Ray Image Detection

To evaluate the performance of the FAST-YOLO algorithm in multi-object detection tasks, we conducted experiments comparing FAST-YOLO with other mainstream algorithms. A unified dataset and configuration parameters were used throughout the experimental process. The experimental results are shown in Figure 4, where the values annotated within the recognition boxes represent confidence scores. This metric is a crucial indicator for assessing the reliability of this algorithm's ability to detect objects in images. The confidence score can be regarded as a model's assessment of the probability of the presence of a particular object, with values ranging from 0 to 1. Higher confidence values indicate a higher level of certainty in the model's judgment regarding an object's presence.

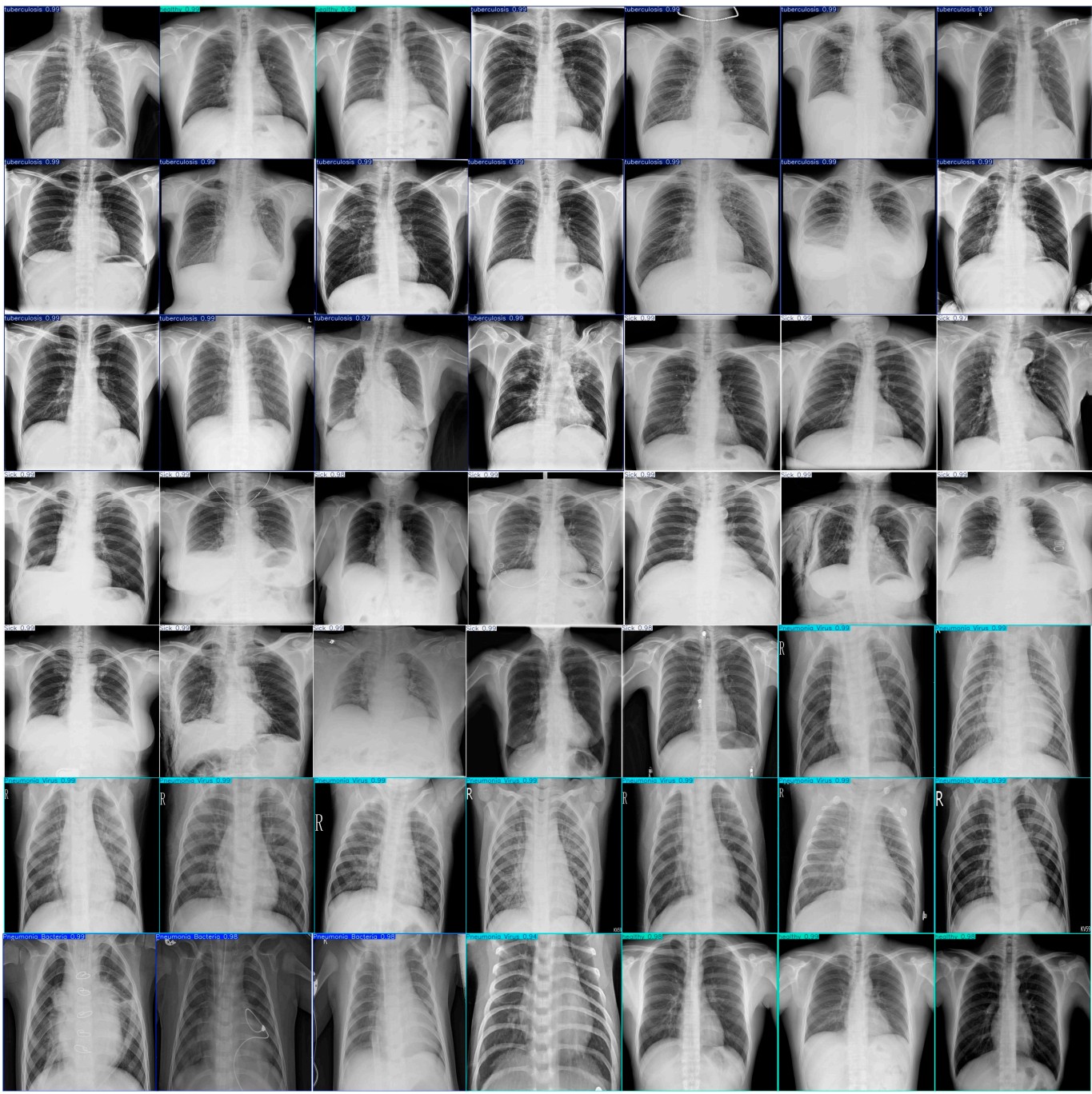

**Figure 4.** Test for pneumonia based on X-ray images.

As illustrated in Figure 5, during the training process, which spanned 500 epochs, the FAST-YOLO model approached convergence at approximately the 80th epoch. Moreover, the precision, recall, and mAP@0.5 values and accuracy are all close to 100%. This demonstrates that the FAST-YOLO model, due to the incorporation of C3k2-DCNV2-DynamicConv, exhibits superior performance in terms of convergence speed as well as precision, recall, mAP@0.5, and mAP@0.5:0.95.

In object detection tasks, each detection result typically requires the assignment of a class label to evaluate a model's classification performance across different categories. After training the FAST-YOLO model, a confusion matrix was generated using the test set to comprehensively assess the model's overall performance. As shown in Figure 6a, the results indicate that the FAST-YOLO's classification performance was satisfactory, providing a

more thorough evaluation of the model's actual performance in object detection tasks, thereby offering strong support for subsequent optimization and improvements.

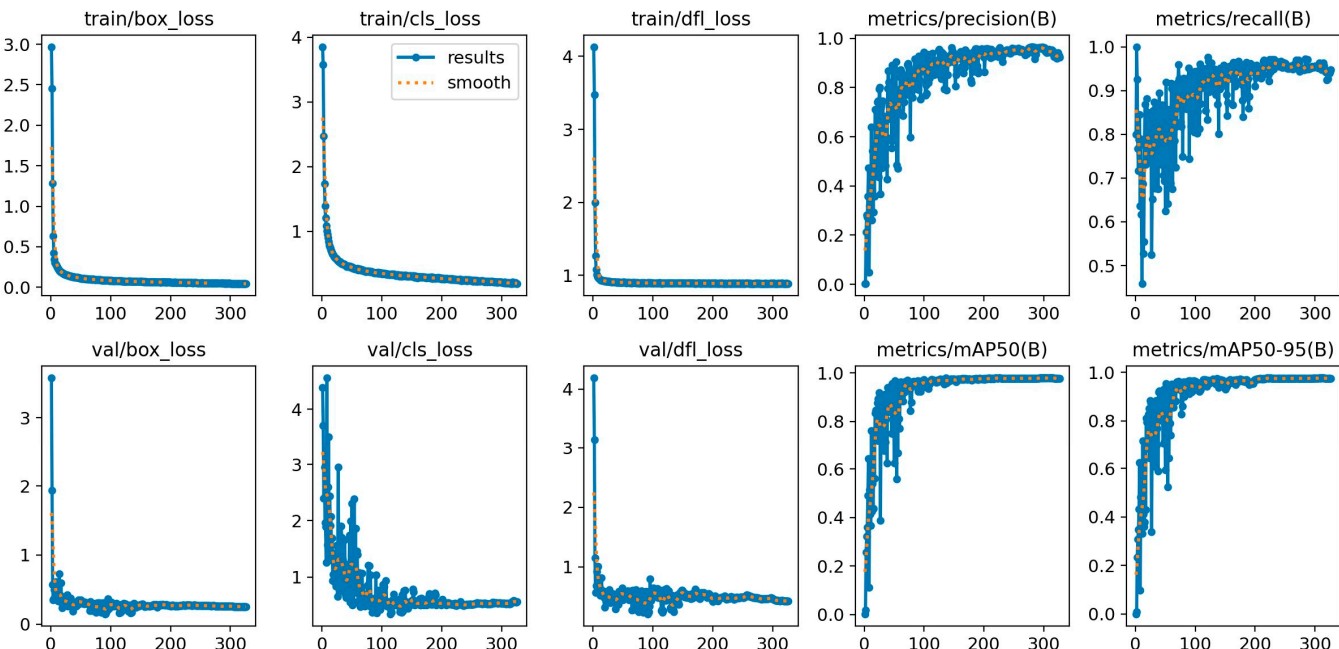

**Figure 5.** FAST-YOLO algorithm training process.

As shown in Figure 6b–e, the performance of the Fast-YOLO model was comprehensively evaluated using performance metrics such as the P-curve, R-curve, F1-curve, and PR curve. These metrics provide multidimensional perspectives for analyzing a model's strengths and weaknesses in different task scenarios, effectively revealing its overall performance characteristics. The P-curve (precision curve) primarily reflects a model's false-positive rate, evaluating its performance in reducing erroneous detections by displaying precision variations at different thresholds. The R-curve (recall curve) reveals a model's false-negative rate, showing its ability to identify true targets in object detection tasks. The F1 curve, based on the weighted harmonic mean of precision and recall, assesses the balance between accuracy and completeness in regard to a model's performance. The PR curve (precision–recall curve) further demonstrates the trade-off between precision and recall at different thresholds, offering a more comprehensive performance evaluation, especially when addressing class imbalance issues. By analyzing these four metrics, the overall performance of the Fast-YOLO model in multiple key dimensions could be assessed. The results show that Fast-YOLO exhibited outstanding performance across various evaluation indicators, confirming its practical application value in complex task environments.

The comparison was conducted using the same dataset and under the same experimental conditions to validate the effectiveness of the FAST-YOLO network in diagnosing pneumonia based on X-ray images and evaluate the proposed algorithm's performance. We trained these networks for 500 epochs and tested the FAST-YOLO network against YOLO series object detection networks, including YOLOv7-Tiny, YOLOv5s, YOLOv5n, YOLOv3-Tiny, and YOLOv3-spp. The test results are shown in Table 3. In terms of precision, recall, and mAP@0.5, the performance of the FAST-YOLO network was similar to that of the other networks. In terms of detection performance, the FAST-YOLO network outperformed the Swin-YOLO, YOLOv11, YOLOv7-Tiny, YOLOv5s, and YOLOv3-spp networks by 66, 48, 53, 39, and 10 FPS, respectively. The mAP@0.5:0.95 value for FAST-YOLO was 2.1%, 14.6%, 17.2%, 18.6%, and 11.2% higher than that for YOLOv7-Tiny, YOLOv5s, YOLOv5n, YOLOv3-Tiny, and YOLOv3-spp, meeting the requirements for diagnosing pneumonia

based on X-ray images. The model effectively balances real-time performance and accuracy demands in medical scenarios. These results demonstrate the innovativeness and value of the FAST-YOLO pneumonia diagnostic network proposed in this study, showing its potential for application to medical devices.

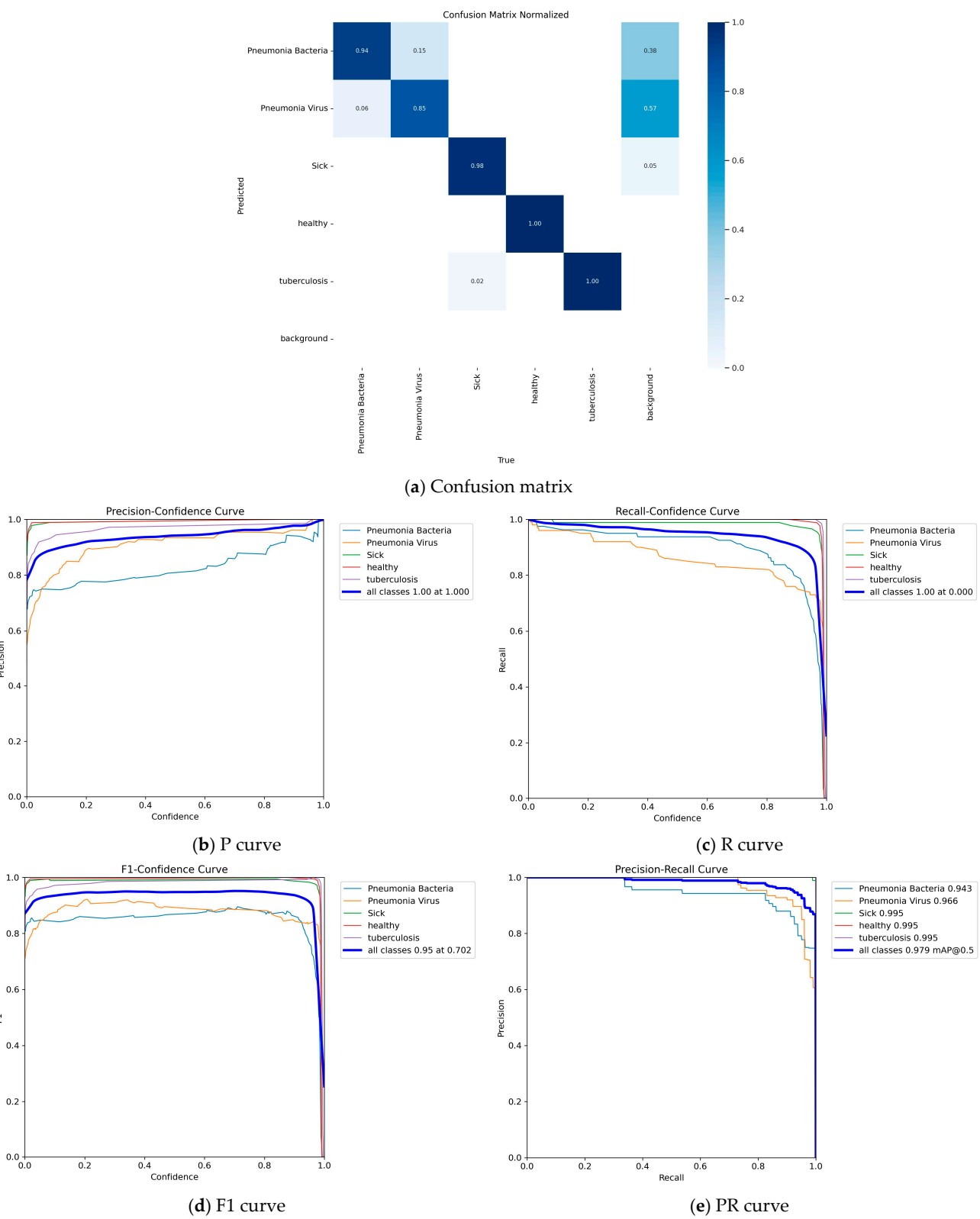

(**a**) Confusion matrix

(**b**) P curve

(**c**) R curve

(**d**) F1 curve

(**e**) PR curve

**Figure 6.** Confusion matrix, P curve, R curve, F1 curve and PR curve performance indicators.

**Table 3.** Performance comparison for the different algorithms.

| Algorithms | Parameter | FPS | Precision | Recall | mAP@0.5 | mAP@0.5:0.95 |
|---|---|---|---|---|---|---|
| YOLOv3-spp | 62,670,264 | 78 | 95.4% | 94.5% | 93.7% | 78.9% |
| YOLOv3-Tiny | 8,711,456 | 94 | 82.2% | 76.2% | 85.3% | 85.0% |
| YOLOv5n | 1,789,624 | 93 | 94.1% | 93.1% | 97.0% | 80.3% |
| YOLOv5s | 7,070,872 | 49 | 95.0% | 96.3% | 96.3% | 82.9% |
| YOLOv7-Tiny | 6,062,584 | 35 | 95.3% | 94.8% | 97.8% | 95.4% |
| YOLOv11 | 2,591,400 | 40 | 94.1% | 95.2% | 97.8% | 97.8% |
| Swin-YOLO | 29,772,626 | 22 | 94.2% | 96.1% | 97.6% | 97.8% |
| FAST-YOLO | 2,279,193 | 88 | 93.4% | 94.7% | 97.7% | 97.5% |

*4.2. Fast-YOLO Generalization Experiment*

The previous experiments demonstrate the excellent performance of the Fast-YOLO algorithm with respect to multi-object workpiece classification datasets. To investigate the generalization capabilities of the Fast-YOLO network further and analyze its detection performance with respect to other publicly available datasets, we employed two open-source datasets for PCB surface defect detection and multi-object workpiece detection for experimentation. As shown in Figure 7, the PCB surface defect detection task involves eight categories: excess solder, missing hole, mouse bite, open circuit, scratch, short circuit, spur, and spurious copper.

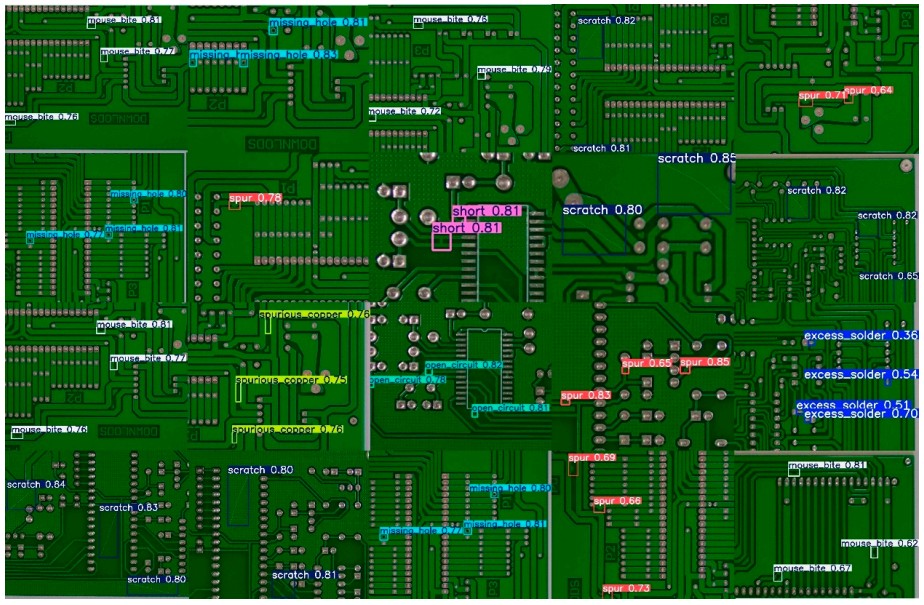

**Figure 7.** Classification detection experiment regarding PCB surface defect detection.

As shown in Table 4, Fast-YOLO's FPS, precision, recall, mAP@0.5, and mAP@0.5:0.95 values in the field of PCB surface defect detection are all superior to those of the other YOLO networks. In PCB board defect detection tasks, there is often a requirement to detect multiple workpieces or defects, which imposes higher demands on the performance of the detection model. The superiority of Fast-YOLO in multi-object detection tasks demonstrates its ability to efficiently perform target separation and precise localization when attempting the detection of multiple targets simultaneously. The exceptional performance of Fast-YOLO makes it highly promising for workpiece defect detection for automated production lines. Fast-YOLO's rapid processing capability and high detection accuracy enable it to monitor the production process in real-time, promptly identifying potential quality issues,

thereby reducing production costs, improving production efficiency, and ensuring the quality of the final products.

**Table 4.** Comparison of the performances of the different algorithms.

| Algorithms | FPS | Precision | Recall | mAP@0.5 | mAP@0.5:0.95 |
|---|---|---|---|---|---|
| YOLOv3-spp | 33 | 90.3% | 82.2% | 85.9% | 46.6% |
| YOLOv3-Tiny | 44 | 76.9% | 58.7% | 67.0% | 26.7% |
| YOLOv5n | 43 | 87.2% | 79.0% | 81.2% | 34.3% |
| YOLOv5s | 31 | 86.6% | 86.4% | 85.2% | 38.5% |
| YOLOv7-Tiny | 34 | 88.7% | 82.1% | 81.9% | 34.7% |
| YOLOv11 | 29 | 91.3% | 90.0% | 92.1% | 45.0% |
| Swin-YOLO | 15 | 92.9% | 91.7% | 94.0% | 46.1% |
| FAST-YOLO | 33 | 93.4% | 91.2% | 94.3% | 47.9% |

As shown in Figure 8, experiments were conducted using a multi-target object classification detection dataset. This dataset, compiled through a combination of laboratory captures and online resources, consists of 14,084 images across 19 categories of multi-target objects.

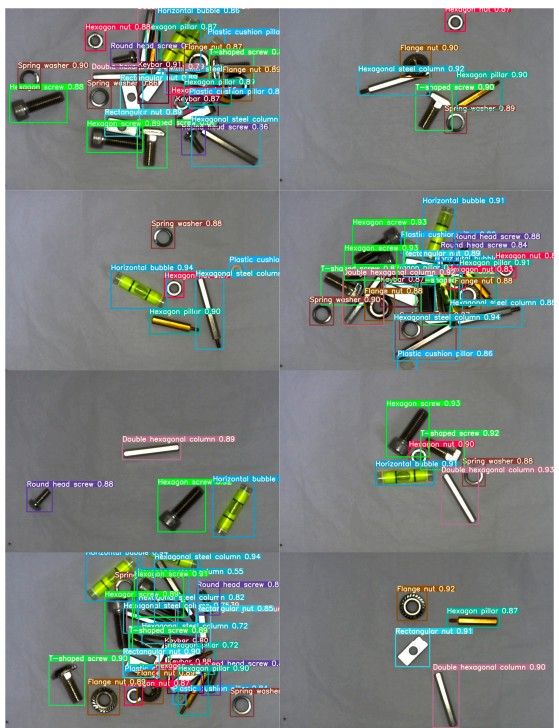

**Figure 8.** Classification detection experiment concerning multi-objective workpiece detection.

According to the detection results in Table 5, Fast-YOLO outperformed other YOLO networks in the multi-target object detection domain, achieving higher FPS, precision, recall, mAP@0.5, and mAP@0.5:0.95. The superior performance of Fast-YOLO in terms of FPS indicates its high computational efficiency in detecting multiple objects, which is critical for real-time detection in industrial applications. In multi-object detection scenarios, where a model must identify and localize multiple targets simultaneously, Fast-YOLO significantly improves processing speed by optimizing the network architecture and inference process while also maintaining accuracy. The outstanding performance of Fast-YOLO makes it highly promising for industrial applications, especially in workpiece classification tasks for automated production lines. Fast-YOLO can simultaneously handle multiple targets in

multi-object workpiece detection tasks, reducing issues such as production line downtime caused by detection delays or false alarms.

**Table 5.** Comparison of the performances of the different algorithms.

| Algorithms | FPS | Precision | Recall | mAP@0.5 | mAP@0.5:0.95 |
|---|---|---|---|---|---|
| YOLOv3-spp | 67 | 98.6% | 98.4% | 99.1% | 82.2% |
| YOLOv3-Tiny | 102 | 95.5% | 83.3% | 90.9% | 65.5% |
| YOLOv5n | 101 | 96.3% | 94.7% | 97.5% | 75.6% |
| YOLOv5s | 50 | 98% | 97.2% | 98.6% | 79.5% |
| YOLOv7-Tiny | 60 | 96.8% | 94.3% | 97.7% | 76.6% |
| YOLOv11 | 81 | 99.3% | 99.2% | 99.3% | 85.0% |
| Swin-YOLO | 38 | 99.2% | 99.1% | 99.3% | 84.3% |
| FAST-YOLO | 94 | 99.2% | 99.5% | 99.4% | 85.3% |

From the above data, it can be found that there were better results for the related evaluation indicators for different data sets, which fully indicates the excellent generalization performance of the Fast-YOLO network.

### 4.3. Deployment of the Diagnostic Network in Practical Applications

In a hospital setting, X-ray images typically involve sensitive patient information. When utilizing the Fast-YOLO model for image detection, it is essential to ensure that the privacy of all data is adequately protected in compliance with relevant laws and regulations. Encryption and anonymization of data during transmission and storage are crucial.

In real-world scenarios, environmental variability may lead to instability or errors in trajectory planning. The Fast-YOLO network model may encounter issues regarding accuracy in practical applications, particularly in cases where lesion types are indistinct or the quality of X-ray images is suboptimal. Furthermore, the generalization capability of the network model may be limited, especially when a hospital's equipment and image quality are not synchronized with the training set. Additionally, sensor data in simulated environments are typically idealized, while robots in real-world operations may face unpredictable external interference or object changes. During the simulation training phase, noise can be introduced into the simulated environment to improve stability. Finally, the differences in inference speed and computational resource requirements between the simulated training environment and the real world may impact the algorithm's real-time performance and effectiveness. Deploying high-performance GPU servers within the hospital or utilizing cloud computing resources to enhance model processing speed can help address this issue. Therefore, transitioning from simulations to the real world typically requires domain adaptation techniques, additional field data, and hardware fine-tuning to overcome these discrepancies.

Doctors may lack sufficient trust in the judgments made by the Fast-YOLO network model, particularly when it comes to medical decision-making. It is important to provide doctors with explanations and rationales for the model's decisions, using visualization tools to reveal the decision-making process of the Fast-YOLO network model, thereby enhancing their understanding of the model's reasoning. In the context of Fast-YOLO network-assisted diagnosis, it is crucial to emphasize the leading role of the doctor, with the Fast-YOLO network serving as a supporting tool designed to improve work efficiency rather than completely replace the doctor. Training and educational programs should be implemented to help doctors understand and accept AI technology.

## 5. Conclusions

In this study, we re-annotated the pneumonia detection YOLO dataset and incorporated augmentation techniques such as Mixup, Mosaic, and Copy–Paste to enhance the discussed model's adaptability to complex scenarios. An automatic diagnostic and detection method for pneumonia X-ray images based on the optimized Fast-YOLO model was proposed. To address the limitations of the traditional YOLO model in detecting small targets and recognizing low-contrast lesions in pneumonia X-ray images, the YOLOv11 model was structurally optimized, and its parameters were adjusted. The Fast-YOLO network improves upon the YOLOv11 architecture by replacing the C3k2 module with the FASPA attention mechanism. This modification effectively retains feature representation capabilities and significantly reduces computational complexity, thereby achieving a balance between computational efficiency and accuracy. The optimized Fast-YOLO network model demonstrated significant advantages in the experiments, achieving high detection accuracy and significantly better processing speed in lesion recognition and localization tasks. The experimental results indicate that, compared to other mainstream object detection models, Fast-YOLO model can achieve comprehensive improvements across key metrics such as FPS, precision, recall, mAP@0.5, and mAP@0.5:0.95, particularly excelling in detection efficiency. These results strongly validate its practical application value in clinical automated diagnosis, meeting the demands for both efficiency and accuracy in clinical practice and holding significant potential for application and further dissemination.

Due to the limited work at present, model interpretability is not deeply discussed in this paper. With the continuous optimization of deep learning model architectures and significant advancements in computational hardware performance, future research on the Fast-YOLO network could integrate large-scale model technologies, global clinical datasets, and multimodal information (such as CT images and patient medical histories). This would further refine detection algorithms, enhancing the accuracy and applicability of automated pneumonia diagnosis systems.

**Author Contributions:** B.Z. and L.C. conceived the experiment(s), L.C. and Z.L. conducted the experiment(s), B.Z. and L.C. analyzed the results. All authors have read and agreed to the published version of the manuscript.

**Funding:** This research was funded by the National Natural Science Foundation of China under Grants (U20A20197), the Provincial Key Research and Development for Liaoning under Grant (2020JH2/10100040). Development and application of autonomous work-ing robots in large scenes (02210073421003). Research and development of automatic inspection flight control technology for satellite denial environment UAV (02210073424000). The key research and development plan of Liaoning Province "Research and develop-ment of multi-scene intelligent robot crowd collaborative command and dispatch system" (2023JH26/10100006).

**Data Availability Statement:** The datasets used and analyzed during the current study are available from the corresponding author on reasonable request.

**Conflicts of Interest:** Author Bin Zhao was employed by the company SIASUN Robot & Automation Co., Ltd. The remaining authors declare that the research was conducted in the absence of any commercial or financial relationships that could be construed as a potential conflict of interest.

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
