# Peer review of "Fast-YOLO Network Model for X-Ray Image Detection of Pneumonia"

_electronics, doi:10.3390/electronics14050903_

Round 1
Reviewer 1 Report
Comments and Suggestions for Authors
1. how does Fast-YOLO compare with other non-YOLO-based architectures, such as transformer-based models or hybrid deep learning approaches?
2. The paper provides a clear explanation of the C3k2, DCNv2, and DynamicConv modules, which enhance the model’s feature extraction capabilities.But could more details be provided about the dataset preprocessing steps?
3. Performance metrics such as precision, recall, mAP@0.5, and FPS are rigorously analyzed, demonstrating the proposed model’s effectiveness.That said, was there any consideration of model interpretability?
4.How can the model’s decisions be explained to clinicians for better trust and adoption?
5. Mathematical formulations of the loss functions and evaluation metrics provide clarity on model optimization.Could a more detailed discussion on why the selected loss functions performed best be included?
6. Would additional fine-tuning be necessary for deployment in different hospitals?
7. That being said, could the readability of certain technical explanations be improved? Some sections (e.g., mathematical formulations) may benefit from more intuitive explanations for non-experts.
Author Response
Dear Reviewers and Editors:
Thank you for your letter and the reviewers’ comments concerning our manuscript entitled ID electronics-3474308. Those comments are all valuable and helpful for revising and improving our paper, as well as the essential guiding significance to our research. We have studied the comments carefully and have made corrections which we hope meet with approval. Revised portions are marked in red in the paper. The significant corrections in the paper and the responses to the reviewer’s comments are as follows: Responds to the reviewer's comments:
Reviewer #1:
1. how does Fast-YOLO compare with other non-YOLO-based architectures, such as transformer-based models or hybrid deep learning approaches?
Response 1: Considering the reviewer’s suggestion, we have added related Swin-transformer experiments in the experiment 4.1 section and 4.2 section.
2. The paper provides a clear explanation of the C3k2, DCNv2, and DynamicConv modules, which enhance the model’s feature extraction capabilities.But could more details be provided about the dataset preprocessing steps?
Response 2: Considering the reviewer’s suggestion and our latest research progress, we have found that FASPA Attention can improve the accuracy and time efficiency of the algorithm more than the C3k2-DCNV2-DynamicConv module. Chapter 3.1 is rewritten, and the structure of FASPA Attention is given. The relative comparison experiments have re-done.
Reviewer 2 required to provide data for download, but our data could not be provided due to confidentiality and hospital requirements, so we replaced the open source data set and re-conducted the experiment.
3. Performance metrics such as precision, recall, mAP@0.5, and FPS are rigorously analyzed, demonstrating the proposed model’s effectiveness. That said, was there any consideration of model interpretability?
Response 3: Dear author, the current research progress is limited, and this question cannot be well answered at present. We have included this opinion in the outlook section of the conclusion and will discuss it in the following article.
4.How can the model’s decisions be explained to clinicians for better trust and adoption?
Response 4: Considering the reviewer’s suggestion, we have added related statements in the experiment 4.3 section.
4.3. Diagnostic network deployment in practical applications
X-ray images in hospitals typically involve sensitive patient information. When utilizing the Fast-YOLO model for image detection, it is essential to ensure that the privacy of all data is adequately protected in compliance with relevant laws and regulations. Encryption and anonymization of data during transmission and storage are crucial.
In real-world scenarios, environmental variability may lead to instability or errors in trajectory planning. The Fast-YOLO network model may encounter issues with accuracy in practical applications, particularly in cases where lesion types are indistinct, or the quality of X-ray images is suboptimal. Furthermore, the generalization capability of the network model may be limited, especially when the hospital's equipment and image quality are not synchronized with the training set. Additionally, sensor data in simulated environments is typically idealized, while robots in real-world operations may face unpredictable external interference or object changes. During the simulation training phase, noise can be introduced into the simulated environment to improve stability. Finally, the differences in inference speed and computational resource requirements between the simulated training environment and the real world may impact the algorithm's real-time performance and effectiveness. Deploying high-performance GPU servers within the hospital or utilizing cloud computing resources to enhance model processing speed can help address this issue. Therefore, transitioning from simulation to the real world typically requires domain adaptation techniques, additional field data, and hardware fine-tuning to overcome these discrepancies.
Doctors may lack sufficient trust in the judgments made by the Fast-YOLO network model, particularly when it comes to medical decision-making. It is important to provide doctors with explanations and rationales for the model's decisions, using visualization tools to display the decision-making process of the Fast-YOLO network model, thereby enhancing their understanding of the model's reasoning. In the context of Fast-YOLO network-assisted diagnosis, it is crucial to emphasize the leading role of the doctor, with the Fast-YOLO network serving as a supporting tool to improve work efficiency rather than completely replacing the doctor. Training and educational programs should be conducted to help doctors understand and accept AI technology.
5. Mathematical formulations of the loss functions and evaluation metrics provide clarity on model optimization.Could a more detailed discussion on why the selected loss functions performed best be included?
Response 5: Considering the reviewer’s suggestion, we have added related statements in the Chapter 2.2 section.
The pneumonia X-ray image detection system not only needs to achieve high accuracy in lesion detection but also should rely on a scientifically designed loss function and evaluation metrics to optimize model performance. In deep learning, the loss function and evaluation metrics are two indispensable core components in model training and evaluation. 1. The loss function defines how the model adjusts its parameters during the training process to minimize prediction errors or maximize a given objective. It directly influences the calculation of gradients and the updating of parameters. The goal of deep learning models is typically to optimize the model's predictive performance by minimizing the loss function. 2. Evaluation metrics are used to assess the model's performance outside of training, helping users understand the model's actual performance across different tasks. Metrics such as Accuracy, Precision, Recall, and F1 Score are often used, especially in cases of class imbalance, as they provide richer information than simple loss functions. The model can be evaluated at different stages of training, and adjustments to hyperparameters or training strategies can be made based on the evaluation metrics, which is a common method for improving model performance. The loss function forms the foundation of the optimization process, directly guiding how the model adjusts its parameters, while evaluation metrics serve as the standard for assessing the model's final performance. Both must work in tandem to achieve optimal performance.
6. Would additional fine-tuning be necessary for deployment in different hospitals?
Response 6: Considering the reviewer’s suggestion, we have added related statements in the experiment 4.3 section.
Question 4 and question 6 are discussed together in section 4.3.
7. That being said, could the readability of certain technical explanations be improved? Some sections (e.g., mathematical formulations) may benefit from more intuitive explanations for non-experts.
Response 7: Considering the reviewer’s suggestion, we have reviewed the whole paper and revised it.
We tried our best to improve the manuscript and made some changes. These changes will not influence the content and framework of the paper. Furthermore, we listed the changes here but marked them in red in the revised paper.
We earnestly appreciate the Editors/Reviewers’ warm work and hope the correction will be approved.
Once again, thank you very much for your comments and suggestions.

Reviewer 2 Report
Comments and Suggestions for Authors This paper proposes an optimized detection model based on FAST-YOLO to improve the recognition accuracy and localization precision of lesions in pneumonia X-ray images. This paper also proposes a targeted online data augmentation method. This method integrates several image enhancement techniques, including Mixup and Mosaic, for comprehensive dataset preprocessing, significantly improving the model's generalization ability and robustness. The subject of this paper is up-to-date. The methodology sound correct. I have the following issues that have to be addressed: 1) "Related program will be open source in the future: " - the scientific paper is not about the promises but facts. Either make the program available now, or do not write about it at all. It is highly recommended to make source codes available to download to make results reproducible. 2) "This study developed a YOLO Pneumonia Detection Dataset by combining experimental imaging with integrated online resources, resulting in 4,194 images. The images are annotated using the Labeling tool (...)" This dataset might be a very useful reference dataset for other researchers. Please consider publishing it online alongside this paper, for example annotations with instruction how to obtain referenced images. You can use github or any other open online repository. 3) Equation (1),(2),(3) what is "1obj"? 4)Figure 4,5 - please make a vector graphic instead of raster. Numbers is Figure 4 are impossible to read. 5) Please expand the discussion of the results from 4.3.Author Response
Dear Reviewers and Editors:
Thank you for your letter and the reviewers’ comments concerning our manuscript entitled ID electronics-3474308. Those comments are all valuable and helpful for revising and improving our paper, as well as the essential guiding significance to our research. We have studied the comments carefully and have made corrections which we hope meet with approval. Revised portions are marked in red in the paper. The significant corrections in the paper and the responses to the reviewer’s comments are as follows: Responds to the reviewer's comments:
Reviewer #2
1) "Related program will be open source in the future: " - the scientific paper is not about the promises but facts. Either make the program available now, or do not write about it at all. It is highly recommended to make source codes available to download to make results reproducible.
Response 1: Considering the reviewer’s suggestion, we have changed to the open source MIMIC-CXR(MIMIC Chest X-ray) dataset, and re-conducted the experiment, and recorded the experimental results according to the open data set.
The open source datasets were not all used, and the number of selected datasets was consistent with the previous datasets in order to verify the effect.
The reasons for changing to open source MIMIC-CXR(MIMIC Chest X-ray) dataset are as follows:
- The relevant information comes from the hospital and requires application.
- Electronics journals do not accept authors from hospitals.
- To avoid disputes, switch to an open source data set.
The address for download:
https://physionet.org/content/mimic-cxr/2.0.0/
2) "This study developed a YOLO Pneumonia Detection Dataset by combining experimental imaging with integrated online resources, resulting in 4,194 images. The images are annotated using the Labeling tool (...)" This dataset might be a very useful reference dataset for other researchers. Please consider publishing it online alongside this paper, for example annotations with instruction how to obtain referenced images. You can use github or any other open online repository.
Response 2: Considering the reviewer’s suggestion, we have have changed to the open source MIMIC-CXR(MIMIC Chest X-ray) dataset, and re-conducted the experiment, and recorded the experimental results according to the open data set.
3) Equation (1),(2),(3) what is "1obj"?
Response 5: Considering the reviewer’s suggestion, we have added related statements.
is an indicator function, which takes the value of 1 when the sample contains the object.
is an indicator function, which takes the value of 1 when the sample does not contain the object.
4)Figure 4,5 - please make a vector graphic instead of raster. Numbers is Figure 4 are impossible to read.
Response 2: Considering the reviewer’s suggestion, we have drawn the picture of experiment. If reviewer still can't read it, refer to the submitted pdf version.
5) Please expand the discussion of the results from 4.3.
Response 5: Considering the reviewer’s suggestion, we have added related statements in the Chapter 4.3 section.
As shown in Table 5, Fast-YOLO's FPS, precision, recall, mAP@0.5 and mAP@0.5:0.95 in the field of PCB surface defect detection are all ahead of other YOLO networks. In PCB board defect detection tasks, there is often a requirement to detect multiple workpieces or defects, which imposes higher demands on the performance of the detection model. The superiority of Fast-YOLO in multi-object detection tasks demonstrates its ability to efficiently perform target separation and precise localization when handling the detection of multiple targets simultaneously. The exceptional performance of Fast-YOLO makes it highly promising for workpiece defect detection in automated production lines. Its rapid processing capability and high detection accuracy enable Fast-YOLO to monitor the production process in real-time, promptly identifying potential quality issues, thereby reducing production costs, improving production efficiency, and ensuring the quality of the final products.
According to the detection results in Table 6, Fast-YOLO outperforms other YOLO networks in the multi-target object detection domain, achieving higher FPS, precision, recall, mAP@0.5, and mAP@0.5:0.95. The superior performance of Fast-YOLO in terms of FPS indicates its high computational efficiency in detecting multiple objects, which is critical for real-time detection in industrial applications. In multi-object detection scenarios, where the model must identify and localize multiple targets simultaneously, Fast-YOLO significantly improves processing speed by optimizing the network architecture and inference process while maintaining accuracy. The outstanding performance of Fast-YOLO makes it highly promising for industrial applications, especially in workpiece classification tasks on automated production lines. Fast-YOLO can simultaneously handle multiple targets in multi-object workpiece detection tasks, reducing issues such as production line downtime caused by detection delays or false alarms.
We tried our best to improve the manuscript and made some changes. These changes will not influence the content and framework of the paper. Furthermore, we listed the changes here but marked them in red in the revised paper.
We earnestly appreciate the Editors/Reviewers’ warm work and hope the correction will be approved.
Once again, thank you very much for your comments and suggestions.

Round 2
Reviewer 2 Report
Comments and Suggestions for Authors
The authors addressed all my remarks. In my opinion, paper can be accepted.